# ON UNIVERSAL EQUIVARIANT SET NETWORKS

**Nimrod Segol & Yaron Lipman**
Department of Computer Science and Applied Mathematics
Weizmann Institute of Science
Rehovot, Israel
{nimrod.segol,yaron.lipman}@weizmann.ac.il

## ABSTRACT

Using deep neural networks that are either invariant or equivariant to permutations in order to learn functions on unordered sets has become prevalent. The most popular, basic models are DeepSets (Zaheer et al., 2017) and PointNet (Qi et al., 2017). While known to be universal for approximating invariant functions, DeepSets and PointNet are not known to be universal when approximating *equivariant* set functions. On the other hand, several recent equivariant set architectures have been proven equivariant universal (Sannai et al., 2019; Keriven & Peyré, 2019), however these models either use layers that are not permutation equivariant (in the standard sense) and/or use higher order tensor variables which are less practical. There is, therefore, a gap in understanding the universality of popular equivariant set models versus theoretical ones.

In this paper we close this gap by proving that: (i) PointNet is not equivariant universal; and (ii) adding a single linear transmission layer makes PointNet universal. We call this architecture PointNetST and argue it is the simplest permutation equivariant universal model known to date. Another consequence is that DeepSets is universal, and also PointNetSeg, a popular point cloud segmentation network (used e.g., in (Qi et al., 2017)) is universal.

The key theoretical tool used to prove the above results is an explicit characterization of all permutation equivariant polynomial layers. Lastly, we provide numerical experiments validating the theoretical results and comparing different permutation equivariant models.

## 1 INTRODUCTION

Many interesting tasks in machine learning can be described by functions $F$ that take as input a set, $X = (x_1, \ldots, x_n)$, and output some per-element features or values, $F(X) = (F(X)_1, \ldots, F(X)_n)$. *Permutation equivariance* is the property required of $F$ so it is well-defined. Namely, it assures that reshuffling the elements in $X$ and applying $F$ results in the same output, reshuffled in the same manner. For example, if $\tilde{X} = (x_2, x_1, x_3, \ldots, x_n)$ then $F(\tilde{X}) = (F(X)_2, F(X)_1, F(X)_3, \ldots, F(X)_n)$.

Building neural networks that are permutation equivariant *by construction* proved extremely useful in practice. Arguably the most popular models are DeepSets Zaheer et al. (2017) and PointNet Qi et al. (2017). These models enjoy small number of parameters, low memory footprint and computational efficiency along with high empirical expressiveness. Although both DeepSets and PointNet are known to be invariant universal (i.e., can approximate arbitrary invariant continuous functions) they are not known to be equivariant universal (i.e., can approximate arbitrary equivariant continuous functions).

On the other hand, several researchers have suggested theoretical permutation equivariant models and proved they are equivariant universal. Sannai et al. (2019) builds a universal equivariant network by taking $n$ copies of $(n-1)$-invariant networks and combines them with a layer that is not permutation invariant in the standard (above mentioned) sense. Keriven & Peyré (2019) solves a more general problem of building networks that are equivariant universal over arbitrary high order input tensors $\mathbb{R}^{n^d}$ (including graphs); their construction, however, uses higher order tensors as hidden variables

which is of less practical value. Yarotsky (2018) proves that neural networks constructed using a finite set of invariant and equivariant polynomial layers are also equivariant universal, however his network is not explicit (i.e., the polynomials are not characterized for the equivariant case) and also of less practical interest due to the high degree polynomial layers.

In this paper we close the gap between the practical and theoretical permutation equivariant constructions and prove:

**Theorem 1.**

*(i) PointNet is not equivariant universal.*

*(ii) Adding a single linear transmission layer (i.e., $\boldsymbol{X} \mapsto \mathbf{1}\mathbf{1}^T\boldsymbol{X}$) to PointNet makes it equivariant universal.*

*(iii) Using ReLU activation the minimal width required for universal permutation equivariant network satisfies $\omega \leq k_{out} + k_{in} + \binom{n+k_{in}}{k_{in}}$.*

This theorem suggests that, arguably, PointNet with an addition of a single linear layer is the simplest universal equivariant network, able to learn arbitrary continuous equivariant functions of sets. An immediate corollary of this theorem is

**Corollary 1.** *DeepSets and PointNetSeg are universal.*

PointNetSeg is a network used often for point cloud segmentation (e.g., in Qi et al. (2017)). One of the benefit of our result is that it provides a simple characterization of universal equivariant architectures that can be used in the network design process to guarantee universality.

The theoretical tool used for the proof of Theorem 1 is an explicit characterization of the permutation equivariant polynomials over sets of vectors in $\mathbb{R}^k$ using power-sum multi-symmetric polynomials. We prove:

**Theorem 2.** *Let $\boldsymbol{P} : \mathbb{R}^{n \times k} \to \mathbb{R}^{n \times l}$ be a permutation equivariant polynomial map. Then,*

$$\boldsymbol{P}(\boldsymbol{X}) = \sum_{|\alpha| \leq n} \boldsymbol{b}_\alpha \boldsymbol{q}_\alpha^T, \tag{1}$$

*where $\boldsymbol{b}_\alpha = (\boldsymbol{x}_1^\alpha, \dots, \boldsymbol{x}_n^\alpha)^T$, $\boldsymbol{q}_\alpha = (q_{\alpha,1}, \dots, q_{\alpha,l})^T$, where $q_{\alpha,j} = q_{\alpha,j}(s_1, \dots, s_t)$, $t = \binom{n+k}{k}$, are polynomials; $s_j(\boldsymbol{X}) = \sum_{i=1}^n \boldsymbol{x}_i^{\alpha_j}$ are the power-sum multi-symmetric polynomials. On the other hand every polynomial map $\boldsymbol{P}$ satisfying Equation 1 is equivariant.*

This theorem, which extends Proposition 2.27 in Golubitsky & Stewart (2002) to sets of vectors using multivariate polynomials, lends itself to expressing arbitrary equivariant polynomials as a composition of entry-wise continuous functions and a single linear transmission, which in turn facilitates the proof of Theorem 1.

We conclude the paper by numerical experiments validating the theoretical results and testing several permutation equivariant networks for the tasks of set classification and regression.

## 2 PRELIMINARIES

**Equivariant maps.** Vectors $\boldsymbol{x} \in \mathbb{R}^k$ are by default column vectors; $\mathbf{0}, \mathbf{1}$ are the all zero and all one vectors/tensors; $\boldsymbol{e}_i$ is the $i$-th standard basis vector; $\boldsymbol{I}$ is the identity matrix; all dimensions are inferred from context or mentioned explicitly. We represent a set of $n$ vectors in $\mathbb{R}^k$ as a matrix $\boldsymbol{X} \in \mathbb{R}^{n \times k}$ and denote $\boldsymbol{X} = (\boldsymbol{x}_1, \boldsymbol{x}_2, \dots, \boldsymbol{x}_n)^T$, where $\boldsymbol{x}_i \in \mathbb{R}^k$, $i \in [n]$, are the columns of $\boldsymbol{X}$. We denote by $S_n$ the permutation group of $[n]$; its action on $\boldsymbol{X}$ is defined by $\sigma \cdot \boldsymbol{X} = (\boldsymbol{x}_{\sigma^{-1}(1)}, \boldsymbol{x}_{\sigma^{-1}(2)}, \dots, \boldsymbol{x}_{\sigma^{-1}(n)})^T$, $\sigma \in S_n$. That is, $\sigma$ is reshuffling the rows of $\boldsymbol{X}$. The natural class of maps assigning a value or feature vector to every element in an input set is permutation equivariant maps:

**Definition 1.** *A map $\boldsymbol{F} : \mathbb{R}^{n \times k} \to \mathbb{R}^{n \times l}$ satisfying $\boldsymbol{F}(\sigma \cdot \boldsymbol{X}) = \sigma \cdot \boldsymbol{F}(\boldsymbol{X})$ for all $\sigma \in S_n$ and $\boldsymbol{X} \in \mathbb{R}^{n \times d}$ is called permutation equivariant.*

**Power-sum multi-symmetric polynomials.** Given a vector $\boldsymbol{z} = (z_1, \dots, z_n) \in \mathbb{R}^n$ the power-sum symmetric polynomials $s_j(\boldsymbol{z}) = \sum_{i=1}^n z_i^j$, with $j \in [n]$, uniquely characterize $\boldsymbol{z}$ up to permuting

its entries. In other words, for $\boldsymbol{z}, \boldsymbol{y} \in \mathbb{R}^n$ we have $\boldsymbol{y} = \sigma \cdot \boldsymbol{z}$ for some $\sigma \in S_n$ if and only if $s_j(\boldsymbol{y}) = s_j(\boldsymbol{z})$ for all $j \in [n]$. An equivalent property is that every $S_n$ invariant polynomial $p$ can be expressed as a polynomial in the power-sum symmetric polynomials, i.e., $p(\boldsymbol{z}) = q(s_1(\boldsymbol{z}), \ldots, s_n(\boldsymbol{z}))$, see Rydh (2007) Corollary 8.4, Briand (2004) Theorem 3. This fact was previously used in Zaheer et al. (2017) to prove that DeepSets is universal for invariant functions. We extend this result to equivariant functions and the multi-feature (sets of vectors) case.

For a vector $\boldsymbol{x} \in \mathbb{R}^k$ and a multi-index vector $\alpha = (\alpha_1, \ldots, \alpha_k) \in \mathbb{N}^k$ we define $\boldsymbol{x}^\alpha = x_1^{\alpha_1} \cdots x_k^{\alpha_k}$, and $|\alpha| = \sum_{i \in [k]} \alpha_i$. A generalization of the power-sum symmetric polynomials to *matrices* exists and is called power-sum multi-symmetric polynomials, defined with a bit of notation abuse: $s_\alpha(\boldsymbol{X}) = \sum_{i=1}^n \boldsymbol{x}_i^\alpha$, where $\alpha \in \mathbb{N}^k$ is a multi-index satisfying $|\alpha| \leq n$. Note that the number of power-sum multi-symmetric polynomials acting on $\boldsymbol{X} \in \mathbb{R}^{n \times k}$ is $t = \binom{n+k}{k}$. For notation simplicity let $\alpha_1, \ldots, \alpha_t$ be a list of all $\alpha \in \mathbb{N}^k$ with $|\alpha| \leq n$. Then we index the collection of power-sum multi-symmetric polynomials as $s_1, \ldots, s_t$.

Similarly to the vector case the numbers $s_j(\boldsymbol{X})$, $j \in [t]$ characterize $\boldsymbol{X}$ up to permutation of its rows. That is $\boldsymbol{Y} = \sigma \cdot \boldsymbol{X}$ for some $\sigma \in S_n$ iff $s_j(\boldsymbol{Y}) = s_j(\boldsymbol{Y})$ for all $j \in [t]$. Furthermore, every $S_n$ invariant polynomial $p : \mathbb{R}^{n \times k} \to \mathbb{R}$ can be expressed as a polynomial in the power-sum multi-symmetric polynomials (see (Rydh, 2007) corollary 8.4), i.e.,

$$p(\boldsymbol{X}) = q(s_1(\boldsymbol{X}), \ldots, s_t(\boldsymbol{X})), \tag{2}$$

These polynomials were recently used to encode multi-sets in Maron et al. (2019).

## 3 EQUIVARIANT MULTI-SYMMETRIC POLYNOMIAL LAYERS

In this section we develop the main theoretical tool of this paper, namely, a characterization of all permutation equivariant polynomial layers. As far as we know, these layers were not fully characterized before.

Theorem 2 provides an explicit representation of arbitrary permutation equivariant polynomial maps $\boldsymbol{P} : \mathbb{R}^{n \times k} \to \mathbb{R}^{n \times l}$ using the basis of power-sum multi-symmetric polynomials, $s_i(\boldsymbol{X})$. The particular use of power-sum polynomials $s_i(\boldsymbol{X})$ has the advantage it can be encoded efficiently using a neural network: as we will show $s_i(\boldsymbol{X})$ can be approximated using a PointNet with a single linear transmission layer. This allows approximating an arbitrary equivariant polynomial map using PointNet with a single linear transmission layer.

A version of this theorem for vectors instead of matrices (i.e., the case of $k = 1$) appears as Proposition 2.27 in Golubitsky & Stewart (2002); we extend their proof to matrices, which is the relevant scenario for ML applications as it allows working with sets of vectors. For $k = 1$ Theorem 2 reduces to the following form: $\boldsymbol{p}(x)_i = \sum_{a \leq n} p_a(s_1(\boldsymbol{x}), \ldots, s_n(\boldsymbol{x})) x_i^a$ with $s_j(\boldsymbol{x}) = \sum_i x_i^j$. For matrices the monomial $x_i^k$ is replaced by $\boldsymbol{x}_i^\alpha$ for a multi-index $\alpha$ and the power-sum symmetric polynomials are replaced by the power-sum multi-symmetric polynomials.

First, note that it is enough to prove Theorem 1 for $l = 1$ and apply it to every column of $\boldsymbol{P}$. Hence, we deal with a vector of polynomials $\boldsymbol{p} : \mathbb{R}^{n \times k} \to \mathbb{R}^n$ and need to prove it can be expressed as $\boldsymbol{p} = \sum_{|\alpha| \leq n} \boldsymbol{b}_\alpha q_\alpha$, for $S_n$ invariant polynomial $q_\alpha$.

Given a polynomial $p(\boldsymbol{X})$ and the cyclic permutation $\sigma^{-1} = (123 \cdots n)$ the following operation, taking a polynomial to a vector of polynomials, is useful in characterizing equivariant polynomial maps:

$$\lceil p \rceil(\boldsymbol{X}) = \begin{pmatrix} p(\boldsymbol{X}) \\ p(\sigma \cdot \boldsymbol{X}) \\ p(\sigma^2 \cdot \boldsymbol{X}) \\ \vdots \\ p(\sigma^{n-1} \cdot \boldsymbol{X}) \end{pmatrix} \tag{3}$$

Theorem 2 will be proved using the following two lemmas:

**Lemma 1.** *Let $\boldsymbol{p} : \mathbb{R}^{n \times k} \to \mathbb{R}^n$ be an equivariant polynomial map. Then, there exists a polynomial $p : \mathbb{R}^{n \times k} \to \mathbb{R}$, invariant to $S_{n-1}$ (permuting the last $n-1$ rows of $\boldsymbol{X}$) so that $\boldsymbol{p} = \lceil p \rceil$.*

*Proof.* Equivariance of $\boldsymbol{p}$ means that for all $\sigma \in S_n$ it holds that $\sigma \cdot \boldsymbol{p}(\boldsymbol{X}) = \boldsymbol{p}(\sigma \cdot \boldsymbol{X})$

$$\sigma \cdot \boldsymbol{p}(\boldsymbol{X}) = \boldsymbol{p}(\sigma \cdot \boldsymbol{X}). \tag{4}$$

Choosing an arbitrary permutation $\sigma \in \text{stab}(1) < S_n$, namely a permutation satisfying $\sigma(1) = 1$, and observing the first row in Equation 4 we get $p_1(\boldsymbol{X}) = p_1(\sigma \cdot \boldsymbol{X}) = p_1(\boldsymbol{x}_1, \boldsymbol{x}_{\sigma^{-1}(2)}, \dots, \boldsymbol{x}_{\sigma^{-1}(n)})$. Since this is true for all $\sigma \in \text{stab}(1)$ $p_1$ is $S_{n-1}$ invariant. Next, applying $\sigma = (1i)$ to Equation 4 and observing the first row again we get $p_i(\boldsymbol{X}) = p_1(\boldsymbol{x}_i, \dots, \boldsymbol{x}_1, \dots)$. Using the invariance of $p_1$ to $S_{n-1}$ we get $\boldsymbol{p} = \lceil p_1 \rceil$. $\qquad \square$

**Lemma 2.** *Let $p : \mathbb{R}^{n \times k} \to \mathbb{R}$ be a polynomial invariant to $S_{n-1}$ (permuting the last $n-1$ rows of $\boldsymbol{X}$) then*

$$p(\boldsymbol{X}) = \sum_{|\alpha| \leq n} \boldsymbol{x}_1^\alpha q_\alpha(\boldsymbol{X}), \tag{5}$$

*where $q_\alpha$ are $S_n$ invariant.*

*Proof.* Expanding $p$ with respect to $\boldsymbol{x}_1$ we get

$$p(\boldsymbol{X}) = \sum_{|\alpha| \leq m} \boldsymbol{x}_1^\alpha p_\alpha(\boldsymbol{x}_2, \dots, \boldsymbol{x}_n), \tag{6}$$

for some $m \in \mathbb{N}$. We first claim $p_\alpha$ are $S_{n-1}$ invariant. Indeed, note that if $p(\boldsymbol{X}) = p(\boldsymbol{x}_1, \boldsymbol{x}_2, \dots, \boldsymbol{x}_n)$ is $S_{n-1}$ invariant, i.e., invariant to permutations of $\boldsymbol{x}_2, \dots, \boldsymbol{x}_n$, then also its derivatives $\frac{\partial^{|\beta|}}{\partial \boldsymbol{x}_1^\beta} p(\boldsymbol{X})$ are $S_{n-1}$ permutation invariant, for all $\beta \in \mathbb{N}^k$. Taking the derivative $\partial^{|\beta|}/\partial \boldsymbol{x}_1^\beta$ on both sides of Equation 6 we get that $p_\beta$ is $S_{n-1}$ equivariant.

For brevity denote $p = p_\alpha$. Since $p$ is $S_{n-1}$ invariant it can be expressed as a polynomial in the power-sum multi-symmetric polynomials, i.e., $p(\boldsymbol{x}_2, \dots, \boldsymbol{x}_n) = r(s_1(\boldsymbol{x}_2, \dots, \boldsymbol{x}_n), \dots, s_t(\boldsymbol{x}_2, \dots, \boldsymbol{x}_n))$. Note that $s_i(\boldsymbol{x}_2, \dots, \boldsymbol{x}_n) = s_i(\boldsymbol{X}) - \boldsymbol{x}_1^{\alpha_i}$ and therefore

$$p(\boldsymbol{x}_2, \dots, \boldsymbol{x}_n) = r(s_1(\boldsymbol{X}) - \boldsymbol{x}_1^{\alpha_1}, \dots, s_t(\boldsymbol{X}) - \boldsymbol{x}_1^{\alpha_t}).$$

Since $r$ is a polynomial, expanding its monomials in $s_i(\boldsymbol{X})$ and $\boldsymbol{x}_1^\alpha$ shows $p$ can be expressed as $p = \sum_{|\alpha| \leq m'} \boldsymbol{x}_1^\alpha \tilde{p}_\alpha$, where $m' \in \mathbb{N}$, and $\tilde{p}_\alpha$ are $S_n$ invariant (as multiplication of invariant $S_n$ polynomials $s_i(\boldsymbol{X})$). Plugging this in Equation 6 we get Equation 5, possibly with the sum over some $n' > n$. It remains to show $n'$ can be taken to be at-most $n$. This is proved in Corollary 5 in Briand (2004) $\qquad \square$

*Proof.* (*Theorem 2*) Given an equivariant $\boldsymbol{p}$ as above, use Lemma 1 to write $\boldsymbol{p} = \lceil p \rceil$ where $p(\boldsymbol{X})$ is invariant to permuting the last $n-1$ rows of $\boldsymbol{X}$. Use Lemma 2 to write $p(\boldsymbol{X}) = \sum_{|\alpha| \leq n} \boldsymbol{x}_1^\alpha q_\alpha(\boldsymbol{X})$, where $q_\alpha$ are $S_n$ invariant. We get,

$$\boldsymbol{p} = \lceil p \rceil = \sum_{|\alpha| \leq n} \boldsymbol{b}_\alpha q_\alpha.$$

The converse direction is immediate after noting that $\boldsymbol{b}_\alpha$ are equivariant and $q_\alpha$ are invariant. $\qquad \square$

## 4 UNIVERSALITY OF SET EQUIVARIANT NEURAL NETWORKS

We consider equivariant deep neural networks $f : \mathbb{R}^{n \times k_{in}} \to \mathbb{R}^{n \times k_{out}}$,

$$\boldsymbol{F}(\boldsymbol{X}) = \boldsymbol{L}_m \circ \nu \circ \dots \circ \nu \circ \boldsymbol{L}_1(\boldsymbol{X}), \tag{7}$$

where $\boldsymbol{L}_i : \mathbb{R}^{n \times k_i} \to \mathbb{R}^{n \times k_{i+1}}$ are affine equivariant transformations, and $\nu$ is an entry-wise non-linearity (e.g., ReLU). We define the width of the network to be $\omega = \max_i k_i$; note that this definition is different from the one used for standard MLP where the width would be $n\omega$, see e.g., Hanin & Sellke (2017). Zaheer et al. (2017) proved that affine equivariant $\boldsymbol{L}_i$ are of the form

$$\boldsymbol{L}_i(\boldsymbol{X}) = \boldsymbol{X}\boldsymbol{A} + \frac{1}{n}\boldsymbol{1}\boldsymbol{1}^T\boldsymbol{X}\boldsymbol{B} + \boldsymbol{1}\boldsymbol{c}^T, \tag{8}$$

where $\boldsymbol{A}, \boldsymbol{B} \in \mathbb{R}^{k_i \times k_{i+1}}$, and $\boldsymbol{c} \in \mathbb{R}^{k_{i+1}}$ are the layer's trainable parameters; we call the linear transformation $\boldsymbol{X} \mapsto \frac{1}{n} \mathbf{1} \mathbf{1}^T \boldsymbol{X} \boldsymbol{B}$ a linear transmission layer.

We now define the equivariant models considered in this paper: The DeepSets (Zaheer et al., 2017) architecture is Equation 7 with the choice of layers as in Equation 8. Taking $\boldsymbol{B} = 0$ in all layers is the PointNet architecture (Qi et al., 2017). PointNetST is an equivariant model of the form Equation 7 with layers as in Equation 8 where only a single layer $\boldsymbol{L}_i$ has a non-zero $\boldsymbol{B}$. The PointNetSeg (Qi et al., 2017) architecture is PointNet composed with an invariant max layer, namely $\max(\boldsymbol{F}(\boldsymbol{X}))_j = \max_{i \in [n]} \boldsymbol{F}(\boldsymbol{X})_{i,j}$ and then concatenating it with the input $\boldsymbol{X}$, i.e., $[\boldsymbol{X}, \mathbf{1} \max(\boldsymbol{F}(\boldsymbol{X}))]$, and feeding is as input to another PointNet $\boldsymbol{G}$, that is $\boldsymbol{G}([\boldsymbol{X}, \mathbf{1} \max(\boldsymbol{F}(\boldsymbol{X}))])$.

We will prove PointNetST is permutation equivariant universal and therefore arguably the simplest permutation equivariant universal model known to date.

Universality of equivariant deep networks is defined next.

**Definition 2.** *Permutation equivariant universality*[1] *of a model* $\boldsymbol{F} : \mathbb{R}^{n \times k_{in}} \to \mathbb{R}^{n \times k_{out}}$ *means that for every permutation equivariant continuous function* $\boldsymbol{H} : \mathbb{R}^{n \times k_{in}} \to \mathbb{R}^{n \times k_{out}}$ *defined over the cube* $K = [0, 1]^{n \times k_{in}} \subset \mathbb{R}^{n \times k_{in}}$, *and* $\epsilon > 0$ *there exists a choice of* $m$ *(i.e., network depth),* $k_i$ *(i.e., network width) and the trainable parameters of* $\boldsymbol{F}$ *so that* $\|\boldsymbol{H}(\boldsymbol{X}) - \boldsymbol{F}(\boldsymbol{X})\|_\infty < \epsilon$ *for all* $\boldsymbol{X} \in K$.

*Proof. (Theorem 1)* Fact *(i)*, namely that PointNet is not equivariant universal is a consequence of the following simple lemma:

**Lemma 3.** *Let* $\boldsymbol{h} = (h_1, \ldots, h_n)^T : \mathbb{R}^n \to \mathbb{R}^n$ *be the equivariant linear function defined by* $h(\boldsymbol{x}) = \mathbf{1} \mathbf{1}^T \boldsymbol{x}$. *There is no* $f : \mathbb{R} \to \mathbb{R}$ *so that* $|h_i(\boldsymbol{x}) - f(x_i)| < \frac{1}{2}$ *for all* $i \in [n]$ *and* $\boldsymbol{x} \in [0, 1]^n$.

*Proof.* Assume such $f$ exists. Let $\boldsymbol{e}_1 = (1, 0, \ldots, 0)^T \in \mathbb{R}^n$. Then,

$$1 = |h_2(\boldsymbol{e}_1) - h_2(\mathbf{0})| \leq |h_2(\boldsymbol{e}_1) - f(0)| + |f(0) - h_2(\mathbf{0})| < 1$$

reaching a contradiction. $\qquad \square$

To prove *(ii)* we first reduce the problem from the class of all continuous equivariant functions to the class of equivariant polynomials. This is justified by the following lemma.

**Lemma 4.** *Equivariant polynomials* $\boldsymbol{P} : \mathbb{R}^{n \times k_{in}} \to \mathbb{R}^{n \times k_{out}}$ *are dense in the space of continuous equivariant functions* $\boldsymbol{F} : \mathbb{R}^{n \times k_{in}} \to \mathbb{R}^{n \times k_{out}}$ *over the cube* $K$.

*Proof.* Take an arbitrary $\epsilon > 0$. Consider the function $f_{ij} : \mathbb{R}^{n \times k_{in}} \to \mathbb{R}$, which denotes the $(i, j)$-th output entry of $\boldsymbol{F}$. By the Stone-Weierstrass Theorem there exists a polynomial $p_{ij} : \mathbb{R}^{n \times k_{in}} \to \mathbb{R}$ such that $\|f_{ij}(\boldsymbol{X}) - p_{ij}(\boldsymbol{X})\|_\infty \leq \epsilon$ for all $\boldsymbol{X} \in K$. Consider the polynomial map $\boldsymbol{P} : \mathbb{R}^{n \times k_{in}} \to \mathbb{R}^{n \times k_{out}}$ defined by $(\boldsymbol{P})_{ij} = p_{ij}$. $\boldsymbol{P}$ is in general not equivariant. To finish the proof we will symmetrize $\boldsymbol{P}$:

$$\left\| \boldsymbol{F}(\boldsymbol{X}) - \frac{1}{n!} \sum_{\sigma \in S_n} \sigma \cdot \boldsymbol{P}(\sigma^{-1} \cdot \boldsymbol{X}) \right\|_\infty = \left\| \frac{1}{n!} \sum_{\sigma \in S_n} \sigma \cdot \boldsymbol{F}(\sigma^{-1} \cdot \boldsymbol{X}) - \frac{1}{n!} \sum_{\sigma \in S_n} \sigma \cdot \boldsymbol{P}(\sigma^{-1} \cdot \boldsymbol{X}) \right\|_\infty$$

$$= \left\| \frac{1}{n!} \sum_{\sigma \in S_n} \sigma \cdot \left( \boldsymbol{F}(\sigma^{-1} \cdot \boldsymbol{X}) - \boldsymbol{P}(\sigma^{-1} \cdot \boldsymbol{X}) \right) \right\|_\infty$$

$$\leq \frac{1}{n!} \sum_{\sigma \in S_n} \epsilon = \epsilon,$$

where in the first equality we used the fact that $\boldsymbol{F}$ is equivariant. This concludes the proof since $\sum_{\sigma \in S_n} \sigma \cdot \boldsymbol{P}(\sigma^{-1} \cdot \boldsymbol{X})$ is an equivariant polynomial map. $\qquad \square$

Now, according to Theorem 2 an arbitrary equivariant polynomial $\boldsymbol{P} : \mathbb{R}^{n \times k_{in}} \to \mathbb{R}^{n \times k_{out}}$ can be written as $\boldsymbol{P} = \sum_{|\alpha| \leq n} \boldsymbol{b}_\alpha(\boldsymbol{X}) \boldsymbol{q}_\alpha(\boldsymbol{X})^T$, where $\boldsymbol{b}_\alpha(\boldsymbol{X}) = \lceil \boldsymbol{x}_1^\alpha \rceil \in \mathbb{R}^n$ and $\boldsymbol{q}_\alpha =$

---

[1]Or just *equivariant universal* in short.

$(q_{\alpha,1}, \ldots, q_{\alpha,k_{out}}) \in \mathbb{R}^{k_{out}}$ are invariant polynomials. Remember that every $S_n$ invariant polynomial can be expressed as a polynomial in the $t = \binom{n+k_{in}}{k_{in}}$ power-sum multi-symmetric polynomials $s_j(\boldsymbol{X}) = \frac{1}{n} \sum_{i=1}^{n} \boldsymbol{x}_i^{\alpha_j}, j \in [t]$ (we use the $1/n$ normalized version for a bit more simplicity later on). We can therefore write $\boldsymbol{P}$ as composition of three maps:

$$\boldsymbol{P} = \boldsymbol{Q} \circ \boldsymbol{L} \circ \boldsymbol{B}, \tag{9}$$

where $\boldsymbol{B} : \mathbb{R}^{n \times k_{in}} \to \mathbb{R}^{n \times t}$ is defined by

$$\boldsymbol{B}(\boldsymbol{X}) = (\boldsymbol{b}(\boldsymbol{x}_1), \ldots, \boldsymbol{b}(\boldsymbol{x}_n))^T,$$

$\boldsymbol{b}(\boldsymbol{x}) = (\boldsymbol{x}^{\alpha_1}, \ldots, \boldsymbol{x}^{\alpha_t})$; $\boldsymbol{L}$ is defined as in Equation 8 with $\boldsymbol{B} = [\boldsymbol{0}, \boldsymbol{I}]$ and $\boldsymbol{A} = [\boldsymbol{e}_1, \ldots, \boldsymbol{e}_{k_{in}}, \boldsymbol{0}]$, where $\boldsymbol{I} \in \mathbb{R}^{t \times t}$ the identity matrix and $\boldsymbol{e}_i \in \mathbb{R}^t$ represents the standard basis (as-usual). We assume $\alpha_j = \boldsymbol{e}_j \in \mathbb{R}^{k_{in}}$, for $j \in [k_{in}]$. Note that the output of $\boldsymbol{L}$ is of the form

$$\boldsymbol{L}(\boldsymbol{B}(\boldsymbol{X})) = (\boldsymbol{X}, \boldsymbol{1}s_1(\boldsymbol{X}), \boldsymbol{1}s_2(\boldsymbol{X}), \ldots, \boldsymbol{1}s_t(\boldsymbol{X})).$$

Finally, $\boldsymbol{Q} : \mathbb{R}^{n \times (k_{in}+t)} \to \mathbb{R}^{n \times k_{out}}$ is defined by

$$\boldsymbol{Q}(\boldsymbol{X}, \boldsymbol{1}s_1, \ldots, \boldsymbol{1}s_t) = (\boldsymbol{q}(\boldsymbol{x}_1, s_1, \ldots, s_t), \ldots, \boldsymbol{q}(\boldsymbol{x}_n, s_1, \ldots, s_t))^T,$$

and $\boldsymbol{q}(\boldsymbol{x}, s_1, \ldots, s_t) = \sum_{|\alpha| \le n} \boldsymbol{x}^\alpha \boldsymbol{q}_\alpha(s_1, \ldots, s_t)^T$.

The decomposition in Equation 9 of $\boldsymbol{P}$ suggests that replacing $\boldsymbol{Q}, \boldsymbol{B}$ with Multi-Layer Perceptrons (MLPs) would lead to a universal permutation equivariant network consisting of PointNet with a single linear transmission layer, namely PointNetST.

The $\boldsymbol{F}$ approximating $\boldsymbol{P}$ will be defined as

$$\boldsymbol{F} = \boldsymbol{\Psi} \circ \boldsymbol{L} \circ \boldsymbol{\Phi}, \tag{10}$$

where $\boldsymbol{\Phi} : \mathbb{R}^{n \times k_{in}} \to \mathbb{R}^{n \times t}$ and $\boldsymbol{\Psi} : \mathbb{R}^{n \times (t+k_{in})} \to \mathbb{R}^{n \times k_{out}}$ are both of PointNet architecture, namely there exist MLPs $\boldsymbol{\phi} : \mathbb{R}^{k_{in}} \to \mathbb{R}^t$ and $\boldsymbol{\psi} : \mathbb{R}^{t+k_{in}} \to \mathbb{R}^{k_{out}}$ so that $\boldsymbol{\Phi}(\boldsymbol{X}) = (\boldsymbol{\phi}(\boldsymbol{x}_1), \ldots, \boldsymbol{\phi}(\boldsymbol{x}_n))^T$ and $\boldsymbol{\Psi}(\boldsymbol{X}) = (\boldsymbol{\psi}(\boldsymbol{x}_1), \ldots, \boldsymbol{\psi}(\boldsymbol{x}_n))^T$. See Figure 1 for an illustration of $\boldsymbol{F}$.

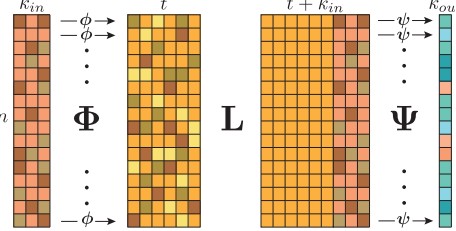

Figure 1: The construction of the universal network (PointNetST).

To build the MLPs $\boldsymbol{\phi}, \boldsymbol{\psi}$ we will first construct $\boldsymbol{\psi}$ to approximate $\boldsymbol{q}$, that is, we use the universality of MLPS (see (Hornik, 1991; Sonoda & Murata, 2017; Hanin & Sellke, 2017)) to construct $\boldsymbol{\psi}$ so that $\|\boldsymbol{\psi}(\boldsymbol{x}, s_1, \ldots, s_t) - \boldsymbol{q}(\boldsymbol{x}, s_1, \ldots, s_t)\|_\infty < \frac{\epsilon}{2}$ for all $(\boldsymbol{x}, s_1, \ldots, s_t) \in [0, 1]^{k_{in}+t}$. Furthermore, as $\boldsymbol{\psi}$ over $[0, 1]^{k_{in}+t}$ is uniformly continuous, let $\delta$ be such that if $\boldsymbol{z}, \boldsymbol{z}' \in [0, 1]^{k_{in}+t}$, $\|\boldsymbol{z} - \boldsymbol{z}'\|_\infty < \delta$ then $\|\boldsymbol{\psi}(\boldsymbol{z}) - \boldsymbol{\psi}(\boldsymbol{z}')\|_\infty < \frac{\epsilon}{2}$. Now, we use universality again to construct $\boldsymbol{\phi}$ approximating $\boldsymbol{b}$, that is we take $\boldsymbol{\phi}$ so that $\|\boldsymbol{\phi}(\boldsymbol{x}) - \boldsymbol{b}(\boldsymbol{x})\|_\infty < \delta$ for all $\boldsymbol{x} \in [0, 1]^{k_{in}}$.

$$\|\boldsymbol{F}(\boldsymbol{X}) - \boldsymbol{P}(\boldsymbol{X})\|_\infty \le \|\boldsymbol{\Psi}(\boldsymbol{L}(\boldsymbol{\Phi}(\boldsymbol{X}))) - \boldsymbol{\Psi}(\boldsymbol{L}(\boldsymbol{B}(\boldsymbol{X})))\|_\infty + \|\boldsymbol{\Psi}(\boldsymbol{L}(\boldsymbol{B}(\boldsymbol{X}))) - \boldsymbol{Q}(\boldsymbol{L}(\boldsymbol{B}(\boldsymbol{X})))\|_\infty$$
$$= \text{err}_1 + \text{err}_2$$

First, $\|\boldsymbol{L}(\boldsymbol{\Phi}(\boldsymbol{X})) - \boldsymbol{L}(\boldsymbol{B}(\boldsymbol{X}))\|_\infty < \delta$ for all $\boldsymbol{X} \in K$ and therefore $\text{err}_1 < \frac{\epsilon}{2}$. Second, note that if $\boldsymbol{X} \in K$ then $\boldsymbol{B}(\boldsymbol{X}) \in [0, 1]^{n \times t}$ and $\boldsymbol{L}(\boldsymbol{B}(\boldsymbol{X})) \in [0, 1]^{n \times (k_{in}+t)}$. Therefore by construction of $\boldsymbol{\psi}$ we have $\text{err}_2 < \frac{\epsilon}{2}$.

To prove *(iii)* we use the result in Hanin & Sellke (2017) (see Theorem 1) bounding the width of an MLP approximating a function $\boldsymbol{f} : [0, 1]^{d_{in}} \to \mathbb{R}^{d_{out}}$ by $d_{in} + d_{out}$. Therefore, the width of the MLP $\boldsymbol{\phi}$ is bounded by $k_{in} + t$, where the width of the MLP $\boldsymbol{\psi}$ is bounded by $t + k_{in} + k_{out}$, proving the bound. $\qquad\square$

We can now prove Cororllary 1.

*Proof. (Corollary 1)*

The fact that the DeepSets model is equivariant universal is immediate. Indeed, The PointNetST model can be obtained from the DeepSets model by setting $\boldsymbol{B} = 0$ in all but one layer, with $\boldsymbol{B}$ as in Equation 8.

For the PointNetSeg model note that by Theorem 1 in Qi et al. (2017) every invariant function $f : \mathbb{R}^{n \times k_{in}} \to \mathbb{R}^t$ can be approximated by a network of the form $\boldsymbol{\Psi}(\mathbf{max}(\boldsymbol{F}(\boldsymbol{X})))$, where $(\boldsymbol{F}(\boldsymbol{X})$ is a PointNet model and $\boldsymbol{\Psi}$ is an MLP. In particular, for every $\varepsilon > 0$ there exists such $\boldsymbol{F}, \boldsymbol{\Phi}$ for which $\|\boldsymbol{\Psi}(\mathbf{max}(\boldsymbol{F}(\boldsymbol{X}))) - (s_1(\boldsymbol{X}), \ldots, s_t(\boldsymbol{X}))\|_\infty < \varepsilon$ for every $\boldsymbol{X} \in [0,1]^{n \times k_{in}}$ where $s_1(\boldsymbol{X}), \ldots, s_t(\boldsymbol{X})$ are the power-sum multi-symmetric polynomials. It follows that we can use PointNetSeg to approximate $\mathbf{1}(s_1(\boldsymbol{X}), \ldots, s_t(\boldsymbol{X}))$. The rest of the proof closely resembles the proof of Theorem 1.

$\square$

**Graph neural networks with constructed adjacency.** One approach sometimes applied to learning from sets of vectors is to define an adjacency matrix (e.g., by thresholding distances of node feature vectors) and apply a graph neural network to the resulting graph (e.g., Wang et al. (2019), Li et al. (2019)). Using the common message passing paradigm (Gilmer et al., 2017) in this case boils to layers of the form: $\boldsymbol{L}(\boldsymbol{X})_i = \psi(\boldsymbol{x}_i, \sum_{j \in N_i} \phi(\boldsymbol{x}_i, \boldsymbol{x}_j)) = \psi(\boldsymbol{x}_i, \sum_{j \in [n]} N(\boldsymbol{x}_i, \boldsymbol{x}_j)\phi(\boldsymbol{x}_i, \boldsymbol{x}_j))$, where $\phi, \psi$ are MLPs, $N_i$ is the index set of neighbors of node $i$, and $N(\boldsymbol{x}_i, \boldsymbol{x}_j)$ is the indicator function for the edge $(i, j)$. If $N$ can be approximated by a continuous function, which is the case at least in the $L_2$ sense for a finite set of vectors, then since $\boldsymbol{L}$ is also equivariant it follows from Theorem 1 that such a network can be approximated (again, at least in $L_2$ norm) arbitrarily well by any universal equivariant network such as PointNetST or DeepSets.

We tested the ability of a DeepSets model with varying depth and width to approximate a single graph convolution layer. We found that a DeepSets model with a small number of layers can approximate a graph convolution layer reasonably well. For details see Appendix A.

## 5 EXPERIMENTS

We conducted experiments in order to validate our theoretical observations. We compared the results of several equivariant models, as well as baseline (full) MLP, on three equivariant learning tasks: a classification task (knapsack) and two regression tasks (squared norm and Fiedler eigen vector). For all tasks we compare results of 7 different models: DeepSets, PointNet, PointNetSeg, PointNetST, PointNetQT and GraphNet. PointNetQT is PointNet with a single quadratic equivariant transmission layer as defined in Appendix B. GraphNet is similar to the graph convolution network in Kipf & Welling (2016) and is defined explicitly in Appendix B. We generated the adjacency matrices for GraphNet by taking 10 nearest neighbors of each set element. In all experiments we used a network of the form Equation 7 with $m = 6$ depth and varying width, fixed across all layers. [2]

**Equivariant classification.** For classification, we chose to learn the multidimensional knapsack problem, which is known to be NP-hard. We are given a set of 4-vectors, represented by $\boldsymbol{X} \in \mathbb{R}^{n \times 4}$, and our goal is to learn the equivariant classification function $f : \mathbb{R}^{n \times 4} \to \{0, 1\}^n$ defined by the following optimization problem:

$$f(\boldsymbol{X}) = \arg\max_z \quad \sum_{i=1}^n x_{i1} z_i$$
$$\text{s.t.} \quad \sum_{i=1}^n x_{ij} z_i \le w_j, \qquad j = 2, 3, 4$$
$$z_i \in \{0, 1\}, \qquad i \in [n]$$

Intuitively, given a set of vectors $\boldsymbol{X} \in \mathbb{R}^{n \times 4}$, $(\boldsymbol{X})_{ij} = x_{ij}$, where each row represents an element in a set, our goal is to find a subset maximizing the value while satisfying budget constraints. The first column of $\boldsymbol{X}$ defines the value of each element, and the three other columns the costs.

To evaluate the success of a trained model we record the percentage of sets for which the predicted subset is such that all the budget constrains are satisfied and the value is within $10\%$ of the optimal value. In Appendix C we detail how we generated this dataset.

**Equivariant regression.** The first equivariant function we considered for regression is the function $f(\boldsymbol{X}) = \mathbf{1} \sum_{i=1}^n \sum_{j=1}^k (\boldsymbol{X}_{i,j} - \frac{1}{2})^2$. Hanin & Sellke (2017) showed this function cannot be approxi-

---

[2]The code can be found at https://github.com/NimrodSegol/On-Universal-Equivariant-Set-Networks

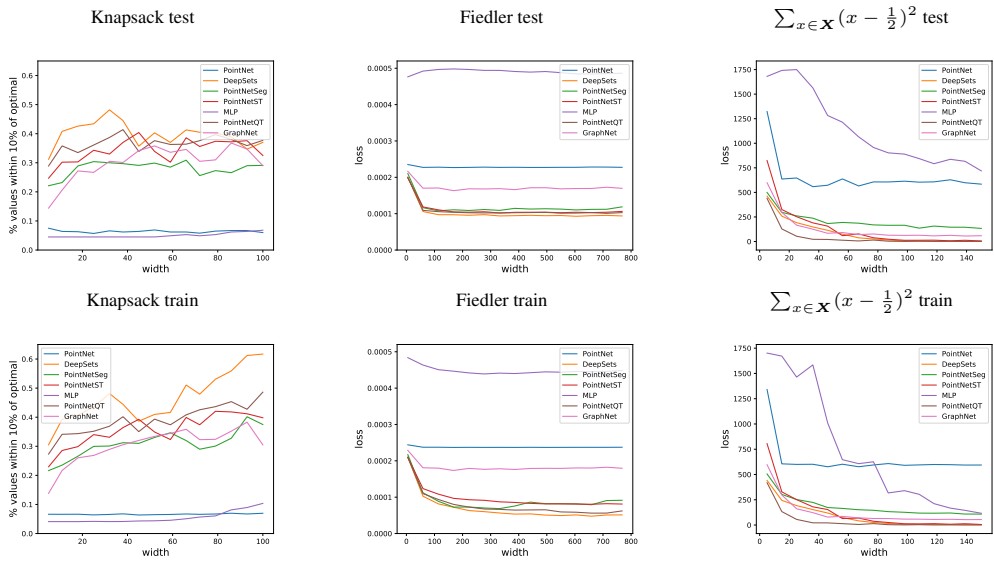

Figure 2: Classification and regression tasks with permutation equivariant models. All the universal permutation equivariant models perform similarly, while the equivariant non-universal PointNet demonstrates reduced performace consistently; MLP baseline (with the same number of parameters as the equivariant models) performs poorly.

mated by MLPs of small width. We drew $10k$ training examples and $1k$ test examples i.i.d. from a $\mathcal{N}(\frac{1}{2}, 1)$ distribution (per entry of $\boldsymbol{X}$).

The second equivariant function we considered is defined on point clouds $\boldsymbol{X} \in \mathbb{R}^{n \times 3}$. For each point cloud we computed a graph by connecting every point to its $10$ nearest neighbors. We then computed the absolute value of the first non trivial eigenvector of the graph Laplacian. We used the ModelNet dataset (Wu et al., 2015) which contains $\sim 9k$ training meshes and $\sim 2k$ test meshes. The point clouds are generated by randomly sampling $512$ points from each mesh.

**Result summary.** Figure 2 summarizes train and test accuracy of the 6 models after training (training details in Appendix C) as a function of the network width $\omega$. We have tested 15 $\omega$ values equidistant in $[5, \frac{n k_{in}}{2}]$.

As can be seen in the graphs, in all three datasets the equivariant universal models (PointNetST, PointNetQT , DeepSets, PointNetSeg) achieved comparable accuracy. PointNet, which is not equivariant universal, consistently achieved inferior performance compared to the universal models, as expected by the theory developed in this paper. The non-equivariant MLP, although universal, used the same width (i.e., same number of parameters) as the equivariant models and was able to over-fit only on one train set (the quadratic function); its performance on the test sets was inferior by a large margin to the equivariant models. We also note that in general the GraphNet model achieved comparable results to the equivariant universal models but was still outperformed by the DeepSets model.

An interesting point is that although the width used in the experiments in much smaller than the bound $k_{out} + k_{in} + \binom{n+k_{in}}{k_{in}}$ established by Theorem 1, the universal models are still able to learn well the functions we tested on. This raises the question of the tightness of this bound, which we leave to future work.

## 6    CONCLUSIONS

In this paper we analyze several set equivariant neural networks and compare their approximation power. We show that while vanilla PointNet (Qi et al., 2017) is not equivariant universal, adding a single linear transmission layer makes it equivariant universal. Our proof strategy is based on a characterization of polynomial equivariant functions. As a corollary we show that the DeepSets model

(Zaheer et al., 2017) and PointNetSeg (Qi et al., 2017) are equivariant universal. Experimentally, we tested the different models on several classification and regression tasks finding that adding a single linear transmitting layer to PointNet makes a significant positive impact on performance.

## 7 ACKNOWLEDGEMENTS

This research was supported in part by the European Research Council (ERC Consolidator Grant, LiftMatch 771136) and the Israel Science Foundation (Grant No. 1830/17).

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

# A    APPROXIMATING GRAPH CONVOLUTION LAYER WITH DEEPSETS

To test the ability of an equivariant universal model to approximate a graph convolution layer, we conducted an experiment where we applied a single graph convolution layer (see Appendix B for a full description of the graph convolution layers used in this paper) with 3 in features and 10 out features. We constructed a knn graph by taking 10 neighbors. We sampled 1000 examples in $\mathbb{R}^{100 \times 3}$ i.i.d from a $\mathcal{N}(\frac{1}{2}, 1)$ distribution (per entry of $X$). The results are summarized in Figure 3. We regressed to the output of a graph convolution layer using the smooth $L_1$ loss.

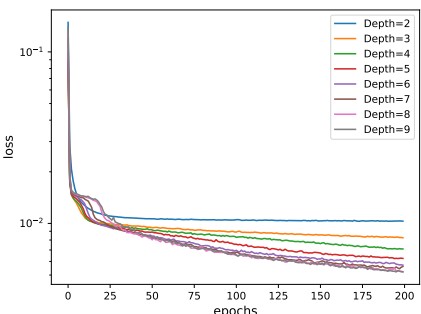 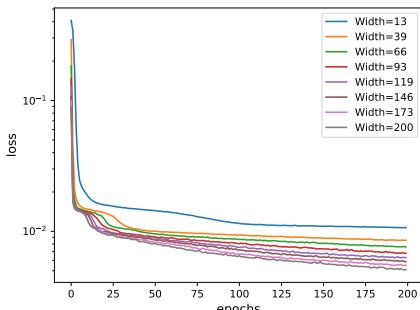

Figure 3: Using DeepSets to regress to a single graph convolution layer. In the left image we varied the depth and took a constant width of 200. In the right we varied the width and took a fixed depth of 6 layers. The $y$ axes are in log scale. Note that even a 2 layer DeepSets can approximate a graph convolution layer up to an error of $\sim 0.01$.

# B    DESCRIPTION OF LAYERS

## B.1    QUADRATIC LAYER

One potential application of Theorem 2 is augmenting an equivariant neural network (Equation 7) with equivariant polynomial layers $P : \mathbb{R}^{n \times k} \to \mathbb{R}^{n \times l}$ of some maximal degree $d$. This can be done in the following way: look for all solutions to $\alpha, \beta_1, \beta_2, \ldots \in \mathbb{N}^k$ so that $|\alpha| + \sum_i |\beta_i| \leq d$. Any solution to this equation will give a basis element of the form $p(X) = \lceil x_1^\alpha \rceil \prod_j \left( \sum_{i=1}^n x_i^{\beta_j} \right)$.

In the paper we tested PointNetQT, an architecture that adds to PointNet a single quadratic equivariant layer. We opted to use only the quadratic transmission operators: For a matrix $X \in \mathbb{R}^{n \times k}$ we define $L(X) \in \mathbb{R}^{n \times k}$ as follows:

$$L(X) = XW_1 + \mathbf{1}\mathbf{1}^T X W_2 + (\mathbf{1}\mathbf{1}^T X) \odot (\mathbf{1}\mathbf{1}^T X) W_3 + (X \odot X) W_4 + (\mathbf{1}\mathbf{1}^T X) \odot X W_5,$$

where $\odot$ is a point-wise multiplication and $W_i \in \mathbb{R}^{n \times k}, i \in [5]$ are the learnable parameters.

## B.2 Graph convolution layer

We implement a graph convolution layers as follows

$$\boldsymbol{L}(\boldsymbol{X}) = \boldsymbol{B}\boldsymbol{X}\boldsymbol{W}_2 + \boldsymbol{X}\boldsymbol{W}_1 + \mathbf{1}\boldsymbol{c}^T$$

with $\boldsymbol{W}_1, \boldsymbol{W}_2, \boldsymbol{c}$ learnable parameters. The matrix $\boldsymbol{B}$ is defined as in Kipf & Welling (2016) from the knn graph of the set. $\boldsymbol{B} = \boldsymbol{D}^{-\frac{1}{2}}\boldsymbol{A}\boldsymbol{D}^{-\frac{1}{2}}$ where $\boldsymbol{D}$ is the degree matrix of the graph and $\boldsymbol{A}$ is the adjacency matrix of the graph with added self-connections.

## C Implementation details

**Knapsack data generation.** We constructed a dataset of $10k$ training examples and $1k$ test examples consisting of $50 \times 4$ matrices. We took $w_1 = 100$, $w_2 = 80$, $w_3 = 50$. To generate $\boldsymbol{X} \in \mathbb{R}^{50 \times 4}$, we draw an integer uniformly at random between 1 and 100 and randomly choose 50 integers between 1 as the first column of $\boldsymbol{X}$. We also randomly chose an integer between 1 and 25 and then randomly chose 150 integers in that range as the three last columns of $\boldsymbol{X}$. The labels for each input $\boldsymbol{X}$ were computed by a standard dynamic programming approach, see Martello & Toth (1990).

**Optimization.** We implemented the experiments in Pytorch Paszke et al. (2017) with the Adam Kingma & Ba (2014) optimizer for learning. For the classification we used the cross entropy loss and trained for 150 epochs with learning rate 0.001, learning rate decay of 0.5 every 100 epochs and batch size 32. For the quadratic function regression we trained for 150 epochs with leaning rate of 0.001, learning rate decay 0.1 every 50 epochs and batch size 64; for the regression to the leading eigen vector we trained for 50 epochs with leaning rate of 0.001 and batch size 32. To regress to the output of a single graph convolution layer we trained for 200 epochs with leaning rate of 0.001 and batch size 32.

