# OpenReview forum: "On Universal Equivariant Set Networks"
_ICLR.cc/2020/Conference — Accept (Poster)_

### Official Review · AnonReviewer3 · 2019-10-21
**Official Blind Review #3**

**Rating:** 6

**Review:**

*CAVEAT*
I must caveat that this paper is out of my comfort zone in terms of topic, so my review below should only be taken lightly. It also explaina the brevity of my review. My apologies to the authors and other reviewers.

*Paper summary*

The authors design a set architecture, which is equivariant to permutations on the input. They show the simplest such set architecture, which preserves equivariance, while being a universal approximator. Nicely this architecture relies on a correction to PointNet, called PointNetST, which they show is not equivariant universal. Furthermore, they run experiments on a few toy examples demonstrating that their system performs well.

*Paper decision*

I have decided to give this paper a weak accept, since it contains both theory and nice experiments. To change to a firm accept, I think the paper needs some changes in written style mainly, to make it friendlier to newcomers to the area, which can easily be implemented in the camera ready stage of paper preparation. For instance, the omission of a results discussion section or a conclusion are clearly not reader friendly.

*Supporting arguments*

- The paper is written clearly. This said, it requires a great deal of effort to follow the maths if you are not already fluent in a lot of the ideas used in the paper (this includes myself).
- I think the structure of the paper is fine for this sort of work. Perhaps at the beginning it would be more useful to spend more time on a roadmap of the results presented in the paper and to explain the exact significance of why the reader should want to continue reading.
- I think the selection of experiments is nice, containing both regression and classification. What would have been nicer would be to perform some sort of ablation study, where the authors studied how the representational capacity of the network changed as a result of them introducing the universal linear transmission layer.
- A direct theoretical and experimental comparison between PointNet and PointNetST would have been useful for me to understand the impact of the change that the authors introduce.

*Questions/notes for the authors*

- Please answer my concerns in the support arguments
- Where is the conclusion section?

**Experience Assessment:**

I have read many papers in this area.

**Review Assessment: Checking Correctness Of Derivations And Theory:**

I did not assess the derivations or theory.

**Review Assessment: Checking Correctness Of Experiments:**

I assessed the sensibility of the experiments.

**Review Assessment: Thoroughness In Paper Reading:**

I made a quick assessment of this paper.

---

> ### Author Response · Authors · 2019-11-07
> **Addressing Reviewer 3 concerns**
>
> We thank the reviewer for the detailed review. Below we address the main concerns.
>
> “I think the paper needs some changes in written style mainly, to make it friendlier to newcomers to the area... For instance, the omission of a results discussion section or a conclusion are clearly not reader friendly”
> >> We discussed the results of the experiments in pages 7-8. We will revise to make this discussion easier to find.
>
>
>
> “What would have been nicer would be to perform some sort of ablation study, where the authors studied how the representational capacity of the network changed as a result of them introducing the universal linear transmission layer.”
> >> We refer the reviewer to Figure 2 where one can see that the PointNet model underperforms across various tasks compared to PointNetST that is identical to PointNet except the addition of a single linear transmission layer. Furthermore, PointNetQT, PointNetSeg, and DeepSets can be seen as different versions of PointNet variations.
>
>
> “A direct theoretical and experimental comparison between PointNet and PointNetST would have been useful for me to understand the impact of the change that the authors introduce..”
> >> In Theorem 1 we state that PointNet is not equivariant universal, but PointNet with a single transmission (PointNetST)  layer is. In the experiments section we compare these two models on three different learning tasks.
>
>
> “Where is the conclusion section?”
> >>We felt it is unnecessary but see we were wrong, we will add one.

---

### Official Review · AnonReviewer1 · 2019-10-24
**Official Blind Review #1**

**Rating:** 6

**Review:**

The paper presents proof that the DeepSets and a variant of PointNet are universal approximators for permutation equivariant functions. The proof uses an expression for equivariant polynomials and the universality of MLP. It then shows that the proposed expression in terms of power-sum polynomials can be constructed in PointNet using a minimal modification to the architecture, or using DeepSets, therefore proving the universality of such deep models.

The results of this paper are important. In terms of presentation, the notation and statement of theorems are precise, however, the presentation is rather dry, and I think the paper can be significantly more accessible. For example, here is an alternative and clearer route presenting the same result: one may study the simple case of having single input channel, for which the output at index "i" of an equivariant polynomial is written as the sum of all powers of input multiplied by a polynomial function of the corresponding power-sum. This second part is indeed what is used in the proof of the universality of the permutation invariant version of DeepSets, making the connection more visible. Generalizing this to the multi-channel input as the next step could make the proof more accessible.

The second issue I would like to raise is related to discussions around the non-universality of the vanilla PointNet model. Given the fact that it applies the same MLP independently to individual set members, it is obvious that it is not universal equivariant (for example, consider a function that performs a fixed permutation to its input), and I fail to see why the paper goes into the trouble of having theorems and experiments just to demonstrate this point. If there were any other objectives beyond this in the experiments could you please clarify?

Finally, could you give a more accurate citation (chapter-page number) for the single-channel version of Theorem 2.?


**Experience Assessment:**

I have published one or two papers in this area.

**Review Assessment: Checking Correctness Of Derivations And Theory:**

I assessed the sensibility of the derivations and theory.

**Review Assessment: Checking Correctness Of Experiments:**

I assessed the sensibility of the experiments.

**Review Assessment: Thoroughness In Paper Reading:**

I read the paper at least twice and used my best judgement in assessing the paper.

---

> ### Author Response · Authors · 2019-11-07
> **Replying to Reviewer 1**
>
> We thank the reviewer for the review and the comments. Below we address the main concerns.
>
> “one may study the simple case of having single input channel, for which the output at index "i" of an equivariant polynomial is written as the sum of all powers of input multiplied by a polynomial function of the corresponding power-sum. This second part is indeed what is used in the proof of the universality of the permutation invariant version of DeepSets, making the connection more visible. Generalizing this to the multi-channel input as the next step could make the proof more accessible ”
> >> We will highlight the connection to the case of single input channel and DeepSets permutation invariance universality.
>
> “The second issue I would like to raise is related to discussions around the non-universality of the vanilla PointNet model. ... if there were any other objectives beyond this in the experiments could you please clarify? “
>
> >> We agree it is trivial (and indeed the proof is a one-liner). If the reviewers feel strongly, we can move it to appendix, however we feel it helps to provide a complete picture. We included it in the experiments as a naive baseline and to show that adding a single transmission layer indeed provides a significant improvement.
>
> “Finally, could you give a more accurate citation (chapter-page number) for the single-channel version of Theorem 2.?“
> >> This result is Proposition 2.27 in Golubitsky&Stuart(2002). We will update the paper to give a more accurate citation.

---

### Official Review · AnonReviewer2 · 2019-10-25
**Official Blind Review #2**

**Rating:** 6

**Review:**

TLDR: The function these deep set networks can approximate is too limited to call these networks universal equivariant set networks. Authors should scope the paper to the specific function family these networks can approximate. No baseline comparison with GraphNets.



The paper proposes theoretical analysis on a set of networks that process features independently through MLPs + global aggregation operations. However, the function of interest is limited to a small family of affine equivariant transformations.

A more general function is

\begin{equation}
P(X)_i = Ax_i + \sum_{j \in N(x_i, X)} B_{(x_j, x_i)} x_j + c
\end{equation}

where $N(x_i, X)$ is the set of index of neighbors within the set $X$. It is trivial to show that this function is permutation equivariant.

Then, can the function family the authors used in the paper approximate this function? No.
Can the proposed permutation equivariant function represent all function the authors used in the paper? Yes.

1) If $B=0$, then the proposed function becomes MLP.
2) If $A=0, N(x_i, X) = [n]$ and $B_{(x_j, x_i)} \leftarrow B$, then this is $\mathbf{1}\mathbf{1}^TXB$, the global aggregation function.

Also, this is the actual function that a lot of people are interested in. Let me go over few more examples.

3) If $N(x_i, X) = $adjacency on a graph and $B_{(x_j, x_i)} \leftarrow B$, then this is a graph neural network "convolution" (it is not a convolution)
Example adjacency $N(x_i, X) = \{j \;| \; \|x_i - x_j\|_p < \delta, x_j \in X\}$.
\begin{equation}
\text{GraphOp}(X)_i = Ax_i + \sum_{j \in \{j \;| \; \|x_i - x_j\|_p < \delta, x_j \in X\}} Bx_j + c
\end{equation}

4) If $x_i = [r,g,b,u,v]$ where $[r,g,b]$ is the color, $[u,v]$ is the pixel coordinate and $N(x_i, X) =$ pixel neighbors within some kernel size, $B(x_j, x_i)$ to be the block diagonal matrix only for the first three dimensions and 0 for the rest, then this is the 2D convolution.


Again, the above function is a more general permutation equivariant function that can represent: a graph neural network layer, a convolution, MLP, global pooling and is one of the most widely used functions in the ML community, not MLP + global aggregation.


Regarding the experiment metrics and plots:

On the Knapsack test, the metric of interest is not the accuracy of individual prediction. Rather, whether the network has successfully predicted the optimal solution, or how close the prediction is to the solution.
For example: success rate within the epsilon radius of the optimal solution while satisfying all the constraints. Fail otherwise. If these networks can truly solve these problems, authors should report the success rate while varying the threshold, not individual accuracy of the items which can be arbitrarily high by violating constraints.

Also, the authors should compare with few more graphnet + transmission layer (GraphNetST) baselines with the graph layers: $P(X)_i = Ax_i + \sum_{j \in N(x_i, X)} Bx_j + c$ and the same single transmission layer $\mathbf{1}\mathbf{1}^TXB$ in PointNetST.
PointNet is a specialization of graphnets and GraphNetST should be added as a baseline with reasonable adjacency.

Also experiment figures are extremely compact. Try using log scale or other lines to make the gaps wider.






Minor

I am quite confused with the name PointNetST. Authors claim adding one layer of DeepSet layer to a PointNet becomes PointNetST, but I see this as a special DeepSet with a single transmission layer. The convention is B -> B', not A + B -> A'. In this case, A: PointNet, B: DeepSet

Lemma 3 is too trivial.

The paper is not very self contained. Spare few lines of equations to define what are DeepSets and PointNetSeg in the paper and point out the difference since these networks are used throughout the paper extensively without proper mathematical definition.

P.2 power sum multi-symmetric polynomials. "For a vector $x \in R^K$ and a multi-index ..." I think it was moved out of the next paragraph since  the same $x$ is defined again as $x \in R^n$ again in the next sentence.
Also, try using the consistent dimension for x throughout the paper, it confuses the reader.



**Experience Assessment:**

I have published one or two papers in this area.

**Review Assessment: Checking Correctness Of Derivations And Theory:**

I assessed the sensibility of the derivations and theory.

**Review Assessment: Checking Correctness Of Experiments:**

I assessed the sensibility of the experiments.

**Review Assessment: Thoroughness In Paper Reading:**

I read the paper at least twice and used my best judgement in assessing the paper.

---

> ### Author Response · Authors · 2019-11-07
> **Addressing Reviewer 2**
>
> We thank the reviewer for the detailed and thoughtful review. We address the reviewers main concerns.
>
> “A more general function is $P(X)_i=Ax_i+\sum_{j\in N(x_i,X)} B(x_j,x_i) x_j + c$, where $N(x_n,X)$ is the set of index of neighbors within the set… Then, can the function family the authors used in the paper approximate this function? No.“
> >> We respectfully disagree. The function P(X) described by the reviewer can be approximated arbitrarily well using a continuous equivariant function (by using bump functions to approximate indicators functions of neighbors). As such it can be approximated with the universal models considered in this paper. (e.g. PointNetST, DeepSets etc.)
> Reiterating the main result of this paper: *Every* continuous equivariant function defined solely on a set of feature vectors can be approximated with PointnetST over a compact domain.
> We are happy to include a discussion about this in the revised paper.
>
>
> “On the Knapsack test, the metric of interest is not the accuracy of individual prediction. Rather, whether the network has successfully predicted the optimal solution, or how close the prediction is to the solution.”
> >> Thank you for this comment. We will revise the Knapsack graph to show the success metric suggested by the reviewer. We remark that the goal here was not to construct a network that solves the Knapsack problem but to demonstrate the difference between universal and non-universal models.
>
> “Spare few lines of equations to define what are DeepSets and PointNetSeg in the paper and point out the difference”
> >> The definition of DeepSets appears in Equations 7 and 8. The PointNetSeg model is described in detail in the discussion before the proof of Corollary 1.  We will make the definitions clearer.
>
>
> “Lemma 3 is too trivial.”
> >> We agree it is trivial (and indeed the proof is a one-liner). If the reviewers feel strongly about this, we can move it to appendix, however we feel it helps to provide a complete picture.
>
> “P.2 power sum multi-symmetric polynomials. "For a vector and a multi-index ..." I think it was moved out of the next paragraph since  the same is defined again as again in the next sentence.”
> >> Thank you. We will make this clear.

---

> > ### Comment · AnonReviewer2 · 2019-11-12
> > **Asking followup questions**
> >
> > Thanks for the clarification.
> >
> > >> "As such it can be approximated with the universal models considered in this paper. (e.g. PointNetST, DeepSets etc.)"
> >
> > I would like to see some experimental results. How many layers of the DeepSet layers would be required to approximate one GraphOp. I am very interested in its capacity to approximate every continuous function and have some realistic estimate of the capacity of these networks.

---

### Author Response · Authors · 2019-11-14
**Revision uploaded:**

We thank the reviewers for their comments. We have uploaded a revision of our paper. The following are the changes made:

Reviewer1: We added an appendix in which we discuss the ability of equivariant universal models to approximate an equivariant graph convolution network. We also added an experiment where we use a DeepSets network to approximate a graph convolution layer.
Reviewer1: A baseline of a graph convolution network based on a knn-graph was added to all of the experiments.
Reviewer1: The metric in the Knapsack experiment was changed to percentage of sets for which the predicted solution satisfies all of the constraints and is within 10% of the optimal solution.
The definition of the DeepSets and PointNetSeg models was edited. We also made slight changes to the proof of Corollary 1 in accordance to the notation in the new definition.
Reviewer2: The preliminaries were revised to include a discussion of the main theorem in the case where $k=1$ and to highlight the connection to the proof of invariant universality that appears in the DeepSets paper [Zaheer et al., 2017].
Reviewer2: A precise reference to the result in Golubitsky&Stuart [2002] (equivatiant polynomials $\mathbb{R}^n \to \mathbb{R}^n$) was added.
Reviewer3: The experimental results are now discussed in a results summary section.
Reviewer3: We added a conclusion section that summarizes our results and contributions.

---

### Decision · Program_Chairs · 2019-12-19

**Decision:**

Accept (Poster)

**Comment:**

This paper shows that DeepSets and PointNet, which are known to be universal for approximating functions, are also universal for approximating equivariant set functions. Reviewer are in agreement that this paper is interesting and makes important contributions. However, they feel the paper could be written to be more accessible.

Based on the reviews and discussions following author response, I recommend accepting this paper. I appreciate the authors for an interesting paper and look forward to seeing it at the conference.